# The Relationship between Flow Experience and Burnout Symptoms: A Systematic Review

**DOI:** 10.3390/ijerph19073865

**Published:** 2022-03-24

**Authors:** Fabienne Aust, Theresa Beneke, Corinna Peifer, Magdalena Wekenborg

**Affiliations:** 1Research Group Work and Health, Department of Psychology, University of Lübeck, 23562 Lübeck, Germany; corinna.peifer@uni-luebeck.de; 2Department of Biological Psychology, Technische Universität Dresden, 01069 Dresden, Germany; theresa.beneke@freenet.de (T.B.); magdalena.wekenborg@tu-dresden.de (M.W.); 3Else Kröner Fresenius Center for Digital Health, Technische Universität Dresden, 01069 Dresden, Germany

**Keywords:** flow experience, burnout symptoms, Flow-Burnout-Model

## Abstract

Background: In today’s performance-oriented society, burnout symptoms, defined as consequences of chronic work stress, are an increasing problem. To counteract this development, the important aims are (1) to find protective and modifiable factors that reduce the risk of developing and harboring burnout symptoms and (2) to understand the underlying mechanisms. A phenomenon potentially furthering both aims is flow experience. Based on the earlier literature, we developed a psycho-physiological “Flow-Burnout-Model”, which postulates positive or negative associations between flow and burnout symptoms, depending on the prevailing situational and personal conditions. Methods: To test our Flow-Burnout-Model, we conducted a systematic literature search encompassing flow and burnout symptoms. Eighteen empirical studies met the inclusion criteria and were analyzed. Results: The findings of the systematic review as a whole suggest a negative association between flow and burnout symptoms, both cross-sectional and longitudinal. According to the findings from longitudinal studies, flow can be interpreted as a protective factor against burnout symptoms, and burnout symptoms can be interpreted as a factor inhibiting flow. In our conclusion, we maintain the assumption of a bidirectional association between flow and burnout symptoms in the Flow-Burnout-Model but modify the initially suggested positive and negative associations between flow and burnout symptoms towards a predominantly negative relationship. Discussion: Mindful of the heterogeneous findings of earlier studies, the resulting comprehensive Flow-Burnout-Model will lay the foundations for future hypothesis-based research. This includes physiological mechanisms explaining the relationship between flow and burnout symptoms, and likewise, the conditions of their longitudinal association.

## 1. Introduction

Two colleagues sit together in the same office, working on similar tasks. Both have been working in the company since their traineeships and have now been with the company for 13 years. Despite all these similarities, one worker is completely absorbed in her tasks and enjoys mastering the challenges, while the other worker experience increasing doubts as to whether he is up to the challenges. He feels emotionally exhausted, shows signs of cynicism, and suffers from reduced personal accomplishment. In other words, while the first worker experiences flow [1] at work, the other worker suffers from burnout symptoms [2]. This example shows that individuals may perceive and approach objectively identical tasks and demands at work in completely different ways. While some people seem to handle challenging situations easily, others use maladaptive coping methods to deal with high job demands and consequently suffer from burnout symptoms [3,4]. But how are these two constructs related, and could they be mutually dependent? To answer this question, the relationship between flow and burnout symptoms will be investigated through a systematic literature review. The results of this systematic review will lay the groundwork for future, hypothesis-based research on the potential role of flow as a factor protecting against burnout emergence and maintenance.

### 1.1. Burnout

Burnout, up to now, is not listed as an independent diagnostic category in the International Classification of Diseases 10 (ICD-10). It is assigned to the residual category Z73—problems related to life-management difficulty [5]. In the ICD-11, it is classified as an occupational phenomenon and defined as “a syndrome conceptualized as resulting from chronic workplace stress that has not been successfully managed” [6]. Following the definition by Maslach et al. [2], the WHO describes three core dimensions of the syndrome: (1) feelings of energy depletion or exhaustion; (2) increased mental distance from one’s job, or feelings of negativism or cynicism related to one’s job; and (3) reduced professional efficacy [6]. In addition to the definition used by the WHO, recent research suggests that additional factors should be considered. For instance, Schaufeli et al. [7] developed the new *Burnout Assessment Tool* that includes, besides exhaustion, further dimensions such as emotional and cognitive impairment, psychological distress, and psychosomatic complaints.

Given the heterogeneous diagnostic criteria, estimated burnout prevalence must be interpreted with caution [8]. However, earlier research reports a high prevalence of burnout symptoms and an increasing trend in prevalence [8,9,10,11]. This trend is exacerbated by the ongoing COVID pandemic [12,13].

This increase in burnout prevalence is especially alarming, in light of the severe consequences of burnout symptoms among them impaired life satisfaction [14], cardiovascular diseases [15], increased suicidal ideation [16], depressive symptoms [17], occupational disability [18], job dissatisfaction and low organizational commitment [19], and lower work performance [20].

Identifying protective factors for burnout symptoms and the underlying mechanism is thus becoming increasingly urgent. The preliminary evidence shows that experienced self-efficacy, for example, can protect against the development of burnout symptoms [21]. As flow is strongly associated with self-efficacy [22], and as it was found to be associated with problem-focused coping [23], the flow experience could be another promising approach toward reducing the risk of developing and continuing to suffer from burnout symptoms.

### 1.2. Flow

Flow is defined as a rewarding state in which one is completely absorbed and can work on an optimally challenging task [1]. Flow can be characterized by the following three core components: absorption, perceived demands-skill balance, and enjoyment during task performance [24]. It is associated with several positive outcomes, such as improved performance [25,26,27,28], well-being [28,29,30], positive affect [31], creativity [32], and physical health [33]. Research shows that it can counteract mental illness symptoms [34]. Besides the well-studied positive effects of flow, there is also evidence of potential undesired effects, such as impaired risk perception or increased risk of becoming addicted to flow-inducing activity [35]. Csíkszentmihályi [36] (p. 70) already postulated that “Flow experience, like everything else, is not *good* in an absolute sense”.

Since flow is primarily experienced in stress-relevant situations (e.g., teaching, [37]; illegal graffiti spraying, [38]), an association between flow experience and psychological and physiological stress seems logical. Peifer and Tan [39] describe the connection between stress and flow by linking the flow channel model by Csíkszentmihályi [1] with the Transactional Stress Model proposed by Lazarus and Folkman [40]. The flow channel model illustrates that flow is experienced if skills and demands balance. Furthermore, boredom occurs if the skills exceed the challenges, and anxiety occurs if the challenges exceed the skills [1]. In line with the Transactional Stress Model, Peifer et al. [41] modified the original flow channel model by adding states they refer to as relaxation and stress. The Transactional Stress Model describes the process of evaluating a situation with the help of various appraisal steps and conveys that individuals experience stress (threat or harm) if the demands of the situation exceed the resources of the person [40]. In this respect, the definitions of anxiety and stress in the corresponding models can be considered equivalent [39]. Flow occurs in a state between relaxation and stress, when demands can be handled positively [39,42]. This notion is in line with Lazarus et al. [43] (p. 209), who describe flow experience as a “powerful sustainer of coping”. However, the physiological pattern during flow suggests that it is a state of (at least moderately) increased physiological arousal [41], which underlines an association with stress and a potential relationship with burnout symptoms.

### 1.3. Relationship between Flow and Burnout Symptoms

Given the association between flow and stress [39,41] as well as the connection of self-efficacy with both flow [22] and burnout symptoms [21], an association between flow and burnout symptoms seems plausible. The present paper aimes to find a theoretically well-founded model that lays the foundations for hypothesis-driven research in the area of flow and burnout symptoms in future research. To achieve this goal, a three-step approach was applied. (1) First, the Transactional Model of Stress and Flow [39] was extended by including burnout symptoms. This newly created Flow-Burnout-Model additionally incorporates relationships between appraisal, physiological patterns, and subjective experiences. Second, (2) the postulated connections between flow and burnout symptoms were examined based on systematic literature analysis. In the final step, (3) the theoretically derived assumptions of the initial Flow-Burnout-Model were aligned with the results of the systematic literature review and, when necessary, adapted.

In summary, the purpose of this review is to develop a comprehensive Flow-Burnout-Model which can be used to generate theoretically sound hypotheses for future research with a special focus on flow as a potential burnout starting point.

## 2. Introducing a Flow-Burnout-Model

The initial Flow-Burnout-Model is based on theoretical considerations. In the following, we present the model in detail by describing its individual components and their interrelationships. The model distinguishes three psychological states: acute stress, relaxation, and flow. It describes the various associations of these states with (1) the appraisal of the situation, (2) physiological arousal, and (3) burnout symptoms.

### 2.1. Appraisal of the Situation

The subjective evaluation of demands and resources or abilities in different situations form the starting point of our initial Flow-Burnout-Model. Hence, we propose that acute stress, relaxation, and flow differ concerning the appraisal of the task demands and the personal abilities necessary to cope with them [1,39,42]. As mentioned above, a perceived imbalance between demands and skills is associated with acute stress (threat or overload, i.e., demands too high related to the person’s abilities) or relaxation (underload, i.e., demands too low related to the person’s abilities). By contrast, flow experience is associated with a perceived match between demands and skills (challenge). Peifer and Tan [39] thus suggest that a potentially stressful situation can be assumed to cause flow experience, acute stress, or relaxation, depending on the subjective appraisal.

### 2.2. Physiological Arousal

Within our initial Flow-Burnout-Model, we propose at least a partial difference between acute stress, relaxation, and flow concerning the physiological states accompanying them. This difference may manifest in the measurable markers of the two systems mainly responsible for the regulation of physiological arousal in humans: (1) the autonomic nervous system (ANS), with its two branches, the sympathetic (SNS) and the parasympathetic nervous system (PNS); and (2) the hypothalamus-pituitary-adrenal (HPA) axis. We, therefore, propose that subjective experiences and physiological arousal are bidirectionally related.

It is well known that acute stress is associated with increased SNS activity and reduced PNS activity [44,45]. Also, concerning the HPA axis, acute stress is associated with increased activity as indicated by an almost linear increase in cortisol secretion with rising levels of acute stress [46,47]. Conversely, the relaxation response is characterized by a decreased activation of the SNS and the HPA axis (decreased cortisol secretion, [46]). In a resting state, the influence of the PNS predominates [44].

Flow experience differs from this pattern of physiological arousal. Even though flow-associated patterns of physiological arousal have been far less examined than acute stress and relaxation, the preliminary research suggests an inverted u-shaped relationship between the flow experience and the activation of the HPA axis, and also the SNS [41,48], implying that during flow a moderate level of physiological arousal occurs. Regarding PNS activity, the associations appear to be more complex, as research has revealed another inverted u-shaped relationship with flow [49] and also generally increased activity of the PNS during flow in stress-relevant contexts [41].

### 2.3. Associations of Stress and Relaxation with Burnout Symptoms

Associations between acute stress, chronic stress, and burnout symptoms are relatively well understood. Exposure to acute stressors comes with increased physiological arousal that can be recovered by an intact PNS [44,45]. However, if stressors occur frequently and/or over a longer period, they can be referred to as chronic stressors which come along with prolonged activity [50]. Chronic stress is defined as persistent demands that threaten to exceed the individual’s resources [51]. This is considered a cause of burnout symptoms [52]; more specifically, chronic work-related stress is a cause of burnout symptoms [53].

Interestingly, burnout symptoms are associated with the elevated perception of acute stress [54], indicating a bi-directional association between acute stress and burnout.

Associations between relaxation and burnout symptoms are less well understood. Research, however, indicates a negative association between relaxation and burnout symptoms, as relaxation interventions effectively combat burnout symptoms [9,55]. Schaufeli [56] reports that the inability to relax could be both an element and a consequence of burnout or an accompanying symptom. Furthermore, daily relaxation had a negative effect, via reduced work-home interference, on emotional exhaustion and cynicism [57].

### 2.4. Associations of Flow with Burnout Symptoms

The direction of the association with burnout symptoms is less straightforward regarding flow experience. In light of theoretical considerations, both positive and negative associations are plausible. On the one hand, in line with the Transactional Model of Stress and Flow [39,42], a negative association between flow and burnout symptoms is likely. Through a cognitive reappraisal of the situation, acute stress is transformed into a flow experience and therefore into a pleasant challenge [39]. As flow is associated with moderate physiological arousal [41], in comparison with acute stress, the arousal decreases due to the reappraisal. The accumulated stress is alleviated long-term, and burnout symptoms should become less likely. Furthermore, opposite associations between flow experience and burnout symptoms with meaningful constructs in the work context, such as employee performance (for an overview, see Peifer & Wolters [20,58]) and job satisfaction [19,59] suggest a negative association between the two variables.

On the other hand, it should be considered that the flow state is accompanied by moderate physiological arousal [41]. In combination with the potentially addictive nature of the flow experience [60], this results in a potentially alarming constellation. In line with the principle of operant conditioning, the pleasant feeling during flow and the positive consequences of flow act as a reward [61]. This means that the activities that caused this condition are more likely to be repeated and behavioral addiction could occur, i.e., pathological gaming or excessive shopping. A consequence could be a loss of control, especially concerning the duration and frequency of the behavioral implementation [61]. In the work context, this could, for example, lead to workaholism, which is “considered as one of the most common addictions that can impact different areas of human functioning” [62] (p. 401) and is defined as an “inner pressures that make the person feel distressed or guilty about not working” [63] (p. 161). In fact, individuals carry out the flow-promoting activities more often and therefore enter a state of physiological activation repeatedly. In the long run, and without sufficient recovery [64], this constant moderate activation could lead to cumulative strain [50]. Chronic stress and ensuing burnout symptoms are possible consequences [52].

In this context, we would like to emphasize the role of recovery: it was shown that a dynamic balance between demands and skills, including regular phases of rest, enhances the likelihood of experiencing flow [64]. This concurs with the findings that individuals who were well recovered in the morning experienced flow more often during the day than individuals who had not recovered [65]. At the same time, studies show that individuals who do not detach from work have a higher risk of developing burnout symptoms (for an overview, see Sonnentag & Fritz [66]). Accordingly, sufficient recovery could serve as a moderator of the effects of flow experience on burnout. Flow could be protective as long as sufficient recovery occurs, but as soon as flow coincides with insufficient recovery due to a loss of control and the development of addictive behavior, it could be conducive to burnout symptoms.

While the theoretical descriptions above suggest that flow can influence burnout symptoms, the causality could also be the other way around: As burnout symptoms are related to depression [17,67], a reduction in energy and a decrease in activity [5] may occur and prevent people from engaging in activities that could potentially lead to flow. Therefore, suffering from burnout symptoms may culminate in a reduced ability to experience flow.

Based on the two explanatory approaches, both positive and negative correlations can occur. Figure 1 illustrates the described potential relationships between appraisal, physiological arousal, subjective experience and burnout symptoms. This review aims to systematize the findings so far on the relationship between flow and burnout symptoms and to identify the cause-effect relationships. Thus, a model can be created to serve as a basis for further research.

## 3. Methods

### 3.1. Search Strategy and Study Eligibility

To identify relevant articles reporting the associations between flow experience and burnout symptoms, a systematic search was carried out in April 2021 using the PubMed, PubPsych, and PsycInfo databases.

The systematic review was conducted with the following search terms (“flow” OR “flow state” OR “flow experience” OR “experience of flow” OR “psychological flow” OR “flow proneness” OR “feeling of being in the zone” OR “optimal experience” OR “absorption” OR “immersion”) AND (“burnout” OR “clinical burnout” OR “occupational burnout” OR “work-related exhaustion” OR “job-related exhaustion” OR “job-related stress” OR “work-related stress”). The search terms included synonyms of flow experience used in relevant articles and synonyms of burnout symptoms used in the systematic review by Rotenstein et al. [8].

The systematic search resulted in 424 papers, and six additional papers were found through an unsystematic search in Google Scholar and a screening of the reference lists of relevant papers. Duplicate entries were removed, resulting in a final sample size of *n* = 358 articles. The abstracts of these articles were scanned by two authors (F.A., T.B.). With the help of the inclusion or exclusion criteria previously specified, the abstracts were checked for eligibility. For the remaining articles, the full text was read and checked again for compliance with the criteria. Discrepancies between the views of these two authors were resolved through discussion.

### 3.2. Inclusion and Exclusion Criteria

Since this systematic review focused on the relationship between flow and burnout symptoms, publications examining direct relationships between these constructs using a dimensional operationalization were considered, likewise papers investigating categorical group differences. Furthermore, only studies with a quantitative, empirical study design were included in the review. Any country of implementation was accepted. The same applies to the context of the study (e.g., work or study). The systematic review included only studies published in English in peer-reviewed scientific journals that were publicly available. These original papers were only considered eligible if they had not been published in the form of a systematic or narrative review, study or review protocol, book or book chapter, meta-analysis, case study, opinion, or “practical guideline”.

Further restrictions were imposed regarding the survey instrument of the burnout symptoms. Based on O’Connor et al. [68], we included the following commonly used and validated inventories: Maslach Burnout Inventory (MBI; [69]), Oldenburg Burnout Inventory (OLBI; [70]), Copenhagen Burnout Inventory (CBI; [71]); The Burnout Measure (BM; [72]), Psychologists’ Burnout Inventory [73], Organisational Social Context Scale [74], and Professional Quality of Life Scale (ProQOL; [75]). Short forms or occupation-specific variants of these measurement instruments, e.g., MBI-Educators Survey (MBI-ES; [76]), were also accepted. No restriction on the operationalization of flow experience was imposed in advance, in order not to further reduce the anticipatedly small number of studies regarding the relationship between flow experience and burnout symptoms. The use of subscales to measure individual sub-dimensions of burnout symptoms or facets of flow experience was declared valid. In contrast, studies on overlapping constructs (i.e., depressive symptoms, chronic fatigue for burnout symptoms and engagement, passion, and relaxation for flow) were excluded.

### 3.3. Data Extraction

Following the screening, information of interest was extracted and tabulated by two of the authors (F.A., T.B.; see Table 1). Some studies examined flow and burnout symptoms in larger and more complex models. For the sake of clarity, the focus was placed on the direct association between the two constructs; some broader associations are briefly outlined. Furthermore, for clarity and comprehensibility, the labels of the various subscales were harmonized and reversed recoded where appropriate (e.g., personal accomplishment was referred to as reduced personal accomplishment with reversed scores).

In addition, a modified version of the Newcastle-Ottawa Quality Assessment Scale (NOS; 8, 111] was used to determine a score for each study to ascertain its quality. The criteria for scoring included the representativeness of the sample, its size, the survey instrument of flow experience and burnout symptoms, and the presentation of descriptive statistics [cf. 8]. A maximum of one point could be awarded for each criterion met, resulting in a maximum score of five. Following the original NOS scale [111], the studies could be classified into three graded quality ranges according to the respective total score: high (four to five points), medium (two to three points), and low (zero to one point) quality. The studies used in the systematic review range between 2.5 and 5 points, the majority having four points. The scores for the NOS are presented in the Appendix A.

## 4. Results

### 4.1. Study Characteristics

The process of identification and selection is depicted in Figure 2.

A detailed overview of the study characteristics is given in Table 1, in alphabetical order. The systematic review includes 18 studies reported in 17 articles.

The studies were published between 2004 and 2020. Concerning study design, 13 of the studies included were cross-sectional, and five were longitudinal.

Overall, 16,521 individuals participated in the studies. The number of participants ranged between 50 and 10,120 per study. With respect to demographic characteristics, 3756 of these participants were female, 2640 male, and five individuals did not report their sex. One study did not report gender percentages of the total sample. Most studies reported a mean age between 29 and 48, whereas in two studies the mean age was under 21, and one study did not report age ranges.

The studies were carried out in eleven different countries, most of them in the Netherlands (*n* = 5), followed by Malaysia (*n* = 3), Germany (*n* = 2), Canada (*n* = 2) and Spain (*n* = 2). One study each was carried out in China, Croatia, France, Italy, Poland, and Sweden. Most studies (*n* = 14) were carried out in the work context, three in an educational setting and one in a more general context. Most studies operationalized burnout symptoms with a version of the MBI (*n* = 13). Thus, either an MBI sum score (*n* = 3) and/or its sub-dimensions (emotional exhaustion (*n* = 10), cynicism (*n* = 5), reduced personal accomplishment (*n* = 5)) were used. Five studies used the OLBI or an adapted version thereof. Here, an OLBI sum score (*n* = 4) and/or its sub-dimensions (exhaustion (*n* = 1), disengagement) were also used. None of the other burnout measurements mentioned above were used in the studies scrutinized.

Flow was predominantly operationalized using the Work-related Flow Inventory (WOLF) or adaptions thereof (*n* = 10) and its facets absorption, enjoyment, and intrinsic motivation, followed by the subscales or single items of the Utrecht Work Engagement Scale (UWES; *n* = 2), the flow experience at work scale (*n* = 2), the Flow Trait Scale-2 (*n* = 2), the Swedish Flow Proneness Questionnaire (*n* = 1) or a combination of the Flow-Short-Scale and a scale developed by Schiefele and Roussakis [105] (*n* = 1).

All details can be found in Table 1.

### 4.2. Relationship between Flow and Burnout Symptoms

In the interests of clarity in the following, the results on the relationship between flow and burnout symptoms are presented systematically, based on (1) cross-sectional vs. longitudinal study designs and (2) the respective burnout measures used. Multifactorial analyses that include different influencing factors are also described.

#### 4.2.1. Associations between Flow and Burnout Symptoms in Cross-Sectional Studies

Most cross-sectional studies used correlative analyses to examine the association between flow and burnout symptoms (*n* = 10). Three studies used other statistical methods to investigate the relationship.

##### Associations between Flow and Burnout Symptoms Measured with the MBI

Two studies used MBI sum scores to operationalize burnout symptoms. Xie et al. [108] used a sum score that included all three MBI sub-dimensions and found a significant negative association with flow (*r* = −0.59) among Chinese medical students (academic burnout, [108]). Flow was operationalized with a sum score of an adapted version of the WOLF, which included the three facets absorption, enjoyment, and intrinsic learning motivation. Martínez-Zaragoza et al. [96], who built an MBI sum score based on the sub-dimension emotional exhaustion and cynicism, found no significant association with professional flow (operationalized as a sum score of the Flow Trait Scale-2 and the MBI sub-dimension personal accomplishment).

Three out of the five studies examining the relationship between flow and the sub-dimension emotional exhaustion, with the help of correlations, found significant negative correlations (*rs* between −0.14 and −0.36; [77,99,103]). Bakker and Geurts [77] used the six-item subscale absorption of the UWES to measure flow, Schiefele et al. [103] used a sum score of an adapted version of the Flow-Short-Scale in combination with a scale developed by Schiefele and Roussakis [105] and Mosing et al. [99] investigated the relationship between flow in different contexts (global, work, leisure, maintenance, and music) and emotional exhaustion using the SFPQ. Significant negative associations were found for the global score and flow at work, leisure, and maintenance, but not for flow in music. On the other hand, two studies found no significant association between flow and emotional exhaustion [89,98]. While Lavigne et al. [89] measured flow with a sum score of the flow experience at work scale with the facets concentration, control, and autotelic experience, Martínez-Zaragoza et al. [98] used a sum score of the Flow Trait Scale-2.

Three of the studies mentioned above additionally examined associations between cynicism, reduced personal accomplishment and flow [89,98,103]. Concerning cynicism, two studies found significant negative associations with flow (*rs* between −0.24 and −0.44; [89,103]). Reduced personal accomplishment was negatively associated with flow in all these three studies (*rs* between −0.24 and −0.60).

Baumgarten et al. [80] used multiple stepwise regression to investigate which factors influenced the three MBI sub-dimensions of burnout. They entered personal and psychosocial factors into the models to explain burnout symptoms, including the flow facets absorption, enjoyment, and intrinsic motivation. Enjoyment was a significant negative predictor for all MBI sub-dimensions (β between −0.19 and −0.30). Absorption only predicted emotional exhaustion symptoms (β = 0.13, *p* = 0.03) significantly positively. Intrinsic motivation did not predict any sub-dimension of burnout symptoms.

##### Associations between Flow and Burnout Symptoms Measured with the OLBI

Only five studies examined the associations between flow and burnout symptoms using the OLBI. Kasa and Hassan [84] and Ljubin-Golub et al. [93] used the sum score of the OLBI, including the sub-dimensions exhaustion and disengagement. While Ljubin-Golub et al. [93] found negative correlations between all flow facets measured with the WOLF-S and the OLBI sum score (absorption: *r* = −0.53, enjoyment: *r* = −0.60, intrinsic motivation: *r* = −0.50), Kasa and Hassan [84] found no significant correlation between flow measured with an adapted version of the WOLF and the OLBI sum score. In a subsequent study, Kasa and Hassan [86] found in a structural equation model that flow was positively associated with the OLBI sum score (β = 1.46), which is in contrast to the findings of other studies.

Zito et al. [109] found negative associations between the OLBI sub-dimension exhaustion and flow, operationalized as the sum score of the WOLF (*r* = −0.54). Kasa and Hassan [87] found that the a-path in their mediation model from burnout symptoms (OLBI sum score) to flow was not significant.

#### 4.2.2. Associations between Flow and Burnout Symptoms in Longitudinal Studies

All longitudinal studies operationalized burnout symptoms using the MBI. Demerouti et al. [82] investigated in a diary study over four days if flow during the working day predicted (1) general emotional exhaustion, (2) exhaustion at work, and (3) exhaustion at home/bedtime. To measure flow, they used three items of each of the flow facets of the WOLF (absorption, enjoyment, intrinsic motivation). All three flow facets, as well as exhaustion at work and exhaustion at home/bedtime, were averaged over the diary days. All three flow facets correlated negatively with general emotional exhaustion (*rs* between −0.12 and −0.43). In contrast, emotional exhaustion at work correlated negatively only with absorption (*r* = −0.14) and enjoyment (*r* = −0.44). Emotional exhaustion at bedtime was negatively associated with enjoyment (*r* = −0.37). Multi-level analysis revealed that emotional exhaustion at work (β = −0.19, *p* < 0.05) and at bedtime (β = −0.17, *p* < 0.05) were only predicted by enjoyment after controlling for nationality, age, presence of children, working hours, and general exhaustion, then again finding negative relationships.

Similar patterns were found in another diary study by Xanthopoulou et al. [107], who collected data over five days. Here, too, flow scores (sum score of an adapted version of WOLF) and emotional exhaustion were aggregated over these five days and were significantly negatively associated (*r* = −0.52). At day-level, the correlation was *r* = −0.43.

Lavigne et al. [89] collected data at two measurement points (T1 and T2) with a time-lag of six months. They used a flow sum score of the flow experience at work scale and the three sub-dimensions of burnout symptoms, measured with the MBI. Flow, measured at T1 and T2, was negatively correlated with emotional exhaustion, cynicism, and reduced personal accomplishment at both measurement points. Correlation coefficients ranged between *r* = −0.42 and *r* = −0.66 for cross-sectional and *r* = −0.39 to *r* = −0.54 for longitudinal associations.

Mäkikangas et al. [95] conducted a study with three measurement points, each with a time-lag of six weeks. Emotional exhaustion, measured only at T1, was negatively associated with flow (operationalized with a sum score of the WOLF) at T1 (*r* = −0.34), T2 (*r* = −0.34), and T3 (*r* = −0.32).

Rodríguez-Sánchez et al. [101] conducted a group comparison. They used the cut-off value of an MBI sum score introduced by Schaufeli et al. [113] to divide the sample into a healthy group and a group suffering from burnout symptoms and collected data via the Experiencing Sampling Method (ESM). The correlation on a personal level (aggregated over 15 days) showed that flow, measured with two items from the UWES, was negatively associated with group affiliation, indicating that burned-out individuals experience less flow (*r* = −0.29). The same pattern was found for the flow facets absorption and enjoyment (*r* = −0.23 and *r* = −0.30). On a timelevel, these results were also found (*rs* between −0.11 and −0.14). Multivariate ANOVA moreover revealed significant group differences regarding the total score for flow (*t* = 8.70, *p* < 0.01), and the facets enjoyment (*t* = 9.62, *p* < 0.05) and absorption (*t* = 5.68, *p* < 0.05). In addition, a multilevel model showed that group affiliation (healthy vs. burned-out) significantly predicted flow experience over time (β = −0.36, *p* < 0.01) when controlling for the effects of time and weekdays. This pattern was also found for the two facets of flow—enjoyment (β = −0.45, *p* < 0.001) and absorption (β = −0.26, *p* < 0.05).

Table 2 provides an overview of how many studies found a positive, negative, or non-significant association between flow and burnout symptoms. Since some studies examined several correlations (e.g., investigating all three flow facets or burnout sub-dimensions) or obtained different results using different analysis methods, the number of studies given does not add up to 18. Most studies (*n* = 6) found a negative relationship between flow and emotional exhaustion. Non-significant results between the same constructs were reported four times. The enjoyment facet showed exclusively negative associations with the various burnout symptoms. In four studies, reduced personal accomplishment was negatively associated with flow.

### 4.3. Multifactorial Associations between Flow and Burnout Symptoms

In addition to reporting relationships between flow and burnout symptoms, other complex explanatory models were investigated, including various influencing factors. An overview is given in Appendix A.

Nine cross-sectional and four longitudinal studies used different models that included flow and burnout symptoms as factors. A serial mediation model used by Ljubin-Golub et al. [93] showed that the relationship between the teacher’s autonomy support and burnout symptoms (OLBI sum score) was mediated by autonomous motivation and study-related flow. Their model with burnout symptoms as a mediator between autonomous motivation and flow did not fit satisfactorily. In another mediation model, it was shown that flow serves as a mediator between job demands/job resources and the OLBI sub-dimension exhaustion [109]. Mäkikangas et al. [95] could find no hypothesized moderating effect of burnout symptoms on the relationship between job resources and flow. In their latent growth curve model, it was shown that the initial level of emotional exhaustion was negatively associated with the initial level of flow and job resources. However, exhaustion did not significantly predict changes in flow experience or job resources over time.

Demerouti et al. [82] found that the interaction between enjoyment and recovery at work predicted emotional exhaustion at work. For participants with low recovery after breaks at work, enjoyment was negatively associated with emotional exhaustion at the end of the working day. For participants with sufficient recovery after breaks, no such relationship was found. Furthermore, the interaction between enjoyment and detachment significantly predicted emotional exhaustion at bedtime. Participants with high levels of enjoyment and detachment scored lower on emotional exhaustion than those with lower levels of detachment. Interactions with absorption and intrinsic motivation were not significant.

In their structural and measurement model describing the relationships between flow, health, burnout, and approach coping, Martínez-Zaragoza et al. [98] found a negative association between flow and reduced personal accomplishment. However, Schiefele et al. [103] used a structural equation model in which burnout symptoms and flow were integrated as outcome variables for dimensions of interest and self-efficacy. In this model, the burnout symptoms and flow were not associated.

Kasa and Hassan [87] found no evidence for flow as a mediator between burnout symptoms and work-family conflicts. Furthermore, they found no mediating effect of flow between burnout symptoms and Organizational Citizenship Behaviour [84]. Socio-culture factors did not moderate the relationship between burnout symptoms and flow [86].

Lavigne et al. [89] used path analyses to test if flow experience mediates the relationship between harmonious passion and the three sub-dimensions of the MBI. They found that flow was a mediator between harmonious passion and cynicism as well as between harmonious passion and reduced personal accomplishment. The mediation model with emotional exhaustion as the outcome was not significant. They also hypothesized that flow acts as a mediator between obsessive passion and the sub-dimensions of the MBI, but these assumptions could not be confirmed. Additionally, they tested a model that used the sub-dimensions of burnout as mediators and flow experience as a dependent variable. This model was found to have a less satisfactory fit. In their longitudinal study, Lavigne et al. [89] reported that harmonious passion measured at T1 was significantly associated with all MBI sub-dimensions at T2. This relationship was mediated by flow experience at T2. Again, these results of mediation could not be found for obsessive passion. It should be noted that non-significant paths were removed in the final model, indicating that flow at T1 did not predict burnout symptoms at T2.

In their multilevel path analysis, Xanthopoulou et al. [107] used deep acting and surface acting as predictors for flow and emotional exhaustion, both of which affected the need for recovery at the end of the working day, relaxation during leisure, and vigor at bedtime. In this complex model, flow and burnout symptoms were significantly negatively associated (*b* = −0.07).

Mosing et al. [99] found in their twin study that the genetic correlation between flow proneness and emotional exhaustion was *r*_g_ = −0.58 and the environmental correlation *r*_e_ = −0.23. When controlling for shared genetic and familial factors, the correlation between flow proneness and emotional exhaustion was still significant (*r* = −0.23).

## 5. Discussion

This article aimed to (1) develop a theoretically based Flow-Burnout-Model, (2) examine the relationship between flow and burnout symptoms with the help of a systematic review, and (3) adapt the initial Flow-Burnout-Model regarding the results of the systematic review.

The initial Burnout-Flow-Model, which is based on the Transactional Model of Stress and Flow [39], presents an explanatory approach to describe the relationship between flow and burnout symptoms, as well as the moderating and mediating factors of this association. In light of the existing literature, we proposed that the appraisal of resources and demands leads to different states of arousal, which are perceived as subjectively different [39]. If the demands exceed the resources, individuals feel stressed [39], which is associated with increased physiological arousal [46]. Conversely, relaxation is experienced as soon as the resources exceed the demands [39]. Physiological arousal is then low [46]. Flow seems to be a state lying between the states of stress and relaxation, at a moderate level of arousal [41]. There is evidence that flow is experienced in stress-relevant situations [1,38] which are perceived as challenges [114].

The relationships between stress, relaxation, and burnout symptoms have been well studied. Burnout symptoms are a consequence of chronic stress [52]. With relaxation, burnout symptoms depict negative associations [9,56,57]. However, the relationship between flow and burnout symptoms is less well understood.

In the following, the results of the systematic review are summarized and discussed. The review considers the empirical evidence of the relationship between flow and burnout symptoms. Based on these findings, the initially proposed Flow-Burnout-Model is adapted and presented.

### 5.1. Discussion of the Systematic Literature Review

In summary, the results of the present systematic review provide convincing evidence for a significant association between flow and burnout symptoms. Hence, the vast majority of studies reported a negative relationship between these two constructs, i.e., increased levels of flow are related to reduced burnout symptoms.

These findings concur with the assumption of the initial Flow-Burnout-Model proposing that enhanced flow experience could serve as a buffer against accumulated acute stress experiences and could, therefore, protect in the long run against burnout symptoms. Furthermore, our review underlines the findings of the existing literature in the field of positive psychology that characterizes flow experience as an important promoter of health and well-being [28,29,30]. The health-promoting effects of flow are at odds with the already demonstrated consequences of burnout symptoms. Additionally, flow is positively associated with positive affect [31] and negatively with negative affect [115], while burnout symptoms are negatively associated with positive affect and positively with negative effect [116]. Therefore, increased positive and decreased negative affect could mediate between flow and burnout symptoms.

Based on our findings, future consideration of the relationship between flow and burnout symptoms seems reasonable.

However, our review also includes studies reporting positive as well as non-significant associations between flow and burnout symptoms; these require closer examination. The positive associations may indicate that flow experience is potentially addictive [60] and could thus lead to a form of workaholism, which in turn is associated positively with burnout symptoms [117]. The non-significant results, on the other hand, could indicate that causality can occur in parallel, their effects thereby possibly canceling each other out. Future research should investigate with the help of mediators, e.g., feeling of addiction, if both effects appear. Differences between the studies regarding sample size and composition, as well as methodological aspects, may account for these mixed results. In the following, their influence on the results revealed will be discussed.

### 5.2. Influence of Additional Variables and Methodology

Overall, the fact that negative correlations between the two constructs were shown across a wide variety of sample compositions concerning age, occupation, sex, nationality, and measurement instruments underlines the robustness of the effects reported.

However, certain variables seem to influence the directionality and extent of these effects. First, two out of three studies that failed to find any significant associations between flow and burnout symptoms examined employees in the Malaysian hotel industry. Accordingly, occupational group and culture may be relevant factors to be considered in future studies. The third study conducted in the Malaysian hotel industry actually found a strong positive association between flow and burnout symptoms, in contrast to the majority of studies, which found a negative association between flow and burnout symptoms. This underlines the assumption that occupational group or culture could be relevant moderators of the association between flow and burnout symptoms. All three studies used the same methodology (OLBI sum score and WOLF sum score) to measure flow and burnout symptoms. To further investigate cultural differences in the relationship between flow and burnout symptoms, more studies should be conducted in countries that differ in their culture and social system, as most studies identified within the present systematic review were conducted in Europe.

Second, our systematic review revealed wide differences in the operationalization of flow and burnout symptoms between the studies included. Flow experience was most often measured with sum scores (*n* = 13), but few studies differentiated between facets of flow (*n* = 4). Only one study reported a sum score and individual facets of flow (*n* = 1). For burnout symptoms, only a few studies used a sum score (*n* = 6), whereas most studies referred to individual burnout sub-dimensions (*n* = 11). Here, too, only one study reported a sum score and individual burnout sub-dimensions (*n* = 1). In addition, one study reported results for three burnout sub-dimensions and three flow facets. More importantly, these differences affected the associations found between flow and burnout symptoms. In the following, we present an overview of the results reliant on the methodology used. Because some studies examined multiple flow facets or burnout sub-dimensions simultaneously and also found both positive, non-significant, or negative associations among the different facets, the results in Table 2 are not limited to the 18 studies reported. In the following compilation of results, some of the findings are further summarized.

#### 5.2.1. Facets of Flow and Burnout Symptoms

In the majority of cases, a negative correlation between flow and burnout symptoms was found (compare Table 2). Nevertheless, the inconsistent results for the facets of flow should be mentioned: In the studies scrutinizing the different facets of flow separately, enjoyment, in particular, showed high negative correlations with burnout symptoms. As every study (*n* = 4) that addressed this facet reported significant negative correlations between enjoyment and burnout symptoms, this finding can be considered very stable. This leads to the assumption that enjoyment, in particular, could be the driving force explaining the protective effect of flow experience on burnout symptoms. This is in line with the interpretation suggested by Baumgarten et al. [80], namely that enjoyment during working hours should be seen as a protective factor against emotional exhaustion and feelings of cynicism. The importance of enjoyment could stem from the fact that, as a positive emotion, it helps to better relieve stressful situations and to show the positive meaning in these situations [118]. In this way acute as well as long-term chronic stress and, consequently, burnout symptoms, would be counteracted.

The results regarding the facet of absorption (*n* = 5) were not entirely consistent. It was found that there were more negative associations with burnout symptoms than positive associations, but there were also some non-significant associations. The negative associations with absorption were found with emotional exhaustion, the MBI-GS and OLBI-S sum score for burnout. Relationships between absorption, reduced personal accomplishment, and cynicism were only reported in one study and were not significant. One study found that absorption was the only flow facet associated with an exacerbation of exhaustion symptoms. This finding substantiates the hypothesis that flow experience has an addictive potential [60] that may eventually result in increased exhaustion. Such reasoning is further supported by the positive correlations between excessive working and exhaustion [80] on the one hand and excessive working and absorption [119] on the other. This permits the assumption that absorption leads to symptoms of emotional exhaustion via workaholism.

The facet of intrinsic motivation was rarely considered in the studies (*n* = 3). Two negative associations emerged (OLBI-S sum score and general emotional exhaustion). On the other hand, associations between intrinsic motivation and emotional exhaustion at work/bedtime and associations between intrinsic motivation and the three MBI sub-dimensions were not significant. This may suggest on the one hand that this facet of flow is less strongly related to burnout symptoms, or, on the other hand, that intrinsic motivation could be positively and negatively associated with burnout symptoms with different mechanisms of action that cancel each other out. More research is needed to investigate the different various possible effects.

Overall, the enjoyment facet is most consistently associated with burnout symptoms, as there are only findings of a negative relationship. Nor can associations be found as consistently for intrinsic motivation and absorption. There were some non-significant associations for intrinsic motivation with burnout symptoms in addition to the few negative correlations. For absorption, there were a few negative associations, but non-significant results were also found, along with one positive correlation. Based on these results, we suggest that the various facets may affect burnout symptoms differently and should also be explored separately in future research. Nevertheless, it should be noted that flow experience is a state in which absorption, enjoyment, and intrinsic motivation coincide [81], and an exclusive focus on individual facets may not adequately cover interrelationships with the overall construct. Accordingly, the overall flow score should always be considered while investigating the associations with burnout symptoms.

#### 5.2.2. Sub-Dimensions of Burnout

As described above, burnout symptoms were measured in many different ways. Most studies focused on the sub-dimension of emotional exhaustion (*n* = 10) while a few reported the sub-dimension of cynicism (*n* = 5) and also reduced personal accomplishment (*n* = 5). There was no obvious pattern whereby certain sub-dimensions were more closely related to flow as effect sizes varied between studies. However, as soon as reduced personal accomplishment was measured, negative correlations emerged with at least one facet of flow, which indicates a very constant relationship. For cynicism, one out of five studies found no significant relationship, while for emotional exhaustion, which was the most frequently examined, two out of ten studies found no relationship to flow at all.

Table 2 indicates that, especially regarding emotional exhaustion, non-significant associations with flow were frequently reported. On the one hand, this could be because emotional exhaustion is the burnout sub-dimension that has been most frequently studied. On the other hand, it could indicate that the association between flow and emotional exhaustion is not as consistent as with other burnout sub-dimensions.

Nevertheless, four studies showed a negative association between cynicism and flow, and eight studies revealed a negative association between emotional exhaustion and at least one facet of flow. Furthermore, there was a significantly negative association between exhaustion (measured with the OLBI) and flow. Accordingly, it can be assumed that the various sub-dimensions of burnout are all related to flow or facets of flow.

### 5.3. Causal Relationship between Flow and Burnout Symptoms

To examine the causal direction of the effect, the results of the longitudinal studies (*n* = 5) merit attention. Both the interval of a few hours [82] and several months between the measurement of flow experience and burnout symptoms [89] (study 2) showed that more frequent flow experience led to lower burnout symptoms. The results permit the assumption that flow experience has a protective effect against burnout symptoms. If future studies confirm the durability of flow-associated reduction in burnout symptoms, setting up flow-promoting workplaces could be a promising approach. As flow experience can be promoted by creating conditions conducive to flow (for an overview, see Peifer & Wolters, [120]), the construct provides a promising approach to proactively prevent burnout symptoms. Since interventions to increase flow experience could be applied at different levels, both the individual and the organization can increase the flow experience at work (e.g., setting clear goals, coaching from the supervisor, [120]). Flow experience can only occur under certain conditions [121]. In the future, studies should investigate to what extent flow can be seen as a protective factor against burnout symptoms or whether accompanying circumstances (e.g., clear goals, unambiguous feedback, perceived demand-skill balance, [121]) exert these protective effects of flow.

In their longitudinal study, Mäkikangas et al. [95] open up a different perspective. They suggest that people with higher emotional exhaustion symptoms experience less flow. Thus, the previously assumed perspective on exhaustion as a depleted state could come into focus, as individuals suffering from burnout symptoms cannot focus all abilities on work demands and further cannot achieve the flow state. Consequently, the buffering effect of the flow experience would be suppressed by already existing burnout symptoms, such as emotional exhaustion. These results are in line with those of Rodríguez-Sánchez et al. [101], who showed that the respective group affiliation (burned-out vs. healthy) was predictive of flow experience, which suggests a flow-inhibiting effect of burnout symptoms. This finding suggests that individuals with high levels of burnout symptoms cannot experience flow in the way healthy people can. The positive consequences of flow (e.g., well-being and performance [28]) can therefore not occur, which leads to a vicious circle. In contrast, Lavigne et al. [89] tested a model in which burnout symptoms predicted flow. However, their model was less satisfactory than the one that reported the reverse effect (flow predicted burnout symptoms), indicating an unidirectional rather than a bidirectional effect. These results align with the cross-sectional study by Ljubin-Golub et al. [93], who found that the model with burnout symptoms as a mediator between autonomous motivation and flow was not as satisfactory as the model that used flow as a mediator between autonomous motivation and burnout symptoms. These disparate results underline the importance of studies that examine both effects on the same data basis using longitudinal studies.

### 5.4. Further Relationships

Although the focus of the systematic review was on the direct relationship between flow experience and burnout symptoms, several other variables seem to influence this relationship. In the work context, harmonious, i.e., non-obsessive, passion for the job [89] and the presence of work-related resources [109] facilitate flow experience, whereas flow experience becomes less likely with increased work demands [109]. In the educational context, autonomous motivation favors the flow experience [93]. Specific behavior also seems to make a difference in whether or how well flow experience serves as a protective factor. While mental distancing from work content helps, high recovery after breaks during working time has been shown to inhibit the positive effect of enjoyment on alleviating exhaustion [82]. The latter could be attributable to the flow experience being interrupted by breaks, and thus its effect on reducing burnout symptoms is blocked [82]. Another approach is that genetic factors can influence the relationship between flow and burnout symptoms [99]. This could mean there are interindividual differences in the relationship between the two constructs. The protective effect of flow proneness on emotional exhaustion was still significant after controlling for shared genetic and familial factors [99].

Our systematic review shows that research in the area of burnout and flow experience is very diverse and linked to different constructs as many studies investigated the relationship between flow and burnout symptoms in combination with other factors. In this review, the focus was on the relationship between flow and burnout symptoms. However, in the future, other influential factors, such as recovery or passion, should also be considered in more detail.

### 5.5. The Flow-Burnout-Model

Regarding the results of the systematic review, the initial Flow-Burnout-Model can be adapted (see Figure 3):

While the initial model assumes there could be both negative and positive associations between flow and burnout symptoms, the adjusted model assumes mainly negative relationships. Nevertheless, some studies (*n* = 2) found that enhanced flow experience (or facets of flow) is associated with more severe burnout symptoms, and there were partly non-significant results. This indicates that the findings should not yet be interpreted as final and that both possible mechanisms of action should continue to be considered in the future. The assumption of a bidirectional association can be maintained as there is evidence that flow is protective of burnout symptoms and that individuals suffering from burnout symptoms may experience less flow. The resulting model is an extension of the Transactional Model of Stress and Flow [39,42] and can also be linked to the assumptions of the Job Demands-Resources Model (J-DR) introduced by Demerouti et al. [122]. Among others, the extended J-DR model [123] postulates a negative association between motivation and strain. Work engagement, a construct closely related to flow [124,125], is assigned to the motivation dimension. Moreover, exhaustion is assigned to strain in this model. Accordingly, our main finding of a negative relationship between flow and burnout symptoms can be reconciled with the predictions of the J-DR model.

Further studies should investigate the relationship between flow and burnout symptoms. These studies should consider the various sub-dimensions of burnout and the facets of flow as our review indicates that these facets and sub-dimensions may be related to each other in different ways. Only when both constructs are broken down appropriately the underlying mechanisms of action can be fully understood. Therefore, future studies should use validated and commonly used instruments to measure flow and burnout symptoms. We recommend using the WOLF [81] to measure flow. It is a validated instrument specifically adapted to the work context in which burnout symptoms occur and considers different facets of the flow experience. Also, the Flow Frequency Scale [126,127] is recommended as this instrument addresses three facets of flow—absorption, perceived demands-skill balance, and enjoyment—that integrate the competing operationalizations of flow [24]. This measurement instrument focuses on the frequency of flow experience and contains one item eliciting whether all three components of the flow occurred simultaneously. We also recommend using the MBI-GS [79] to measure burnout symptoms as the WHO uses the basis of this questionnaire in the definition for the ICD-11 categorization. Furthermore, the newly developed Burnout Assessment Tool [7] could be used in future research as it covers more aspects of the burnout syndrome and takes account of recent findings regarding the conceptualization of burnout. Combining these instruments would allow accurate conclusions to be drawn about the interrelationships between different flow facets and the burnout sub-dimension, thus providing a sound basis for further scientific research. In the process, sum scores should also be formed in each case to check whether the total values are also interrelated.

In addition, it seems worthwhile to develop long-term study designs that examine the relationship between flow and burnout symptoms over several years. In this way, it would be possible to investigate whether any short-term positive effects of the flow experience can also have negative consequences over time. To date, no study has been found that considers such long periods, so the review reveals a gap in research. Furthermore, experimental settings (e.g., intervention studies) could be used to investigate the relationship between flow and burnout symptoms in more detail.

### 5.6. Limitations and Implications for Future Research

When looking at the studies included in our review, the heterogeneity of the operationalization is noticeable regarding flow experience and burnout symptoms, which, in combination with the small number of studies, makes it difficult to identify a pattern and should therefore be taken into account when interpreting the results.

If the measurement methods used for flow experience are compared, two tendencies emerge that indicate the variability of the operationalization of the construct. First, individual authors have adapted and shortened existing measurement instruments without validating them. Secondly, the core of flow experience is also interpreted differently in different measurement instruments. While the primary contents of the WOLF are absorption, intrinsic motivation, and enjoyment [81], the flow experience at work scale [90] focuses on concentration and control regarding professional tasks. It is therefore debatable whether the measurement instruments actually capture the same construct.

Concerning the burnout construct, this could be due to its different interpretations. Those authors who hold the view that burnout symptoms should be operationalized in terms of a multidimensional concept are opposed by the view that emotional exhaustion as a core component is a sufficient indicator of burnout symptoms [99] and that the decoupling of these individual sub-dimensions from the remaining burnout components is permissible. As this reasoning is frequently found in the literature, studies were also included that only recorded emotional exhaustion. This may have further increased the heterogeneity in the level of associations. The inconsistency of measurement not only reflects the lack of scientific consensus on the definition of burnout symptoms [8] but also points to the limited comparability of the results.

Another limitation is that some studies in the systematic review have not previously hypothesized the relationship between flow and burnout symptoms. The correlations reported are more likely to have arisen as by-products when, for example, correlation tables were reported across all constructs measured. Furthermore, the associations between flow and burnout symptoms are in some studies described in more complex analysis models. In these models, the simple connections between flow and burnout symptoms are not presented; they are influenced by other factors. This could distort the results. With the proposed Flow-Burnout-Model, the foundation for hypothesis-driven research is now laid, and future research can be conducted on this basis.

## 6. Conclusions

Overall, the systematic review constitutes an advance in the research on flow and burnout symptoms. The increasing prevalence of burnout symptoms underlines that the search for protective factors is essential. Through the theoretical derivation of a model linking the flow experience and burnout symptoms, associations and underlying mechanisms could be described, and possible effects of the two constructs on each other could be demonstrated.

Furthermore, the systematic review can provide an overview of the relationships between flow and burnout symptoms. Through the review, it becomes clear that the different facets of flow and the sub-dimensions of burnout are related to each other in different ways. This leads to the recommendation that in future research, these facets and sub-dimensions should be explicitly taken into account to be able to comprehend the individual associations even better. Moreover, other contributory factors should be considered (e.g., recreation, cultural differences, effects of time) as different studies reported various influential factors.

The mainly negative associations found between flow and burnout symptoms in this review play a crucial role in both research and practice. On the one hand, the Flow-Burnout-Model can be used to develop new questions for research and thus expand research into protective factors against burnout. On the other hand, based on our results, flow-promoting interventions at work could be designed for practice as they are negatively associated with burnout symptoms. The positive consequences that would accompany flow (e.g., less emotional exhaustion) could have a positive impact on both employees and employers.

## Figures and Tables

**Figure 1 ijerph-19-03865-f001:**
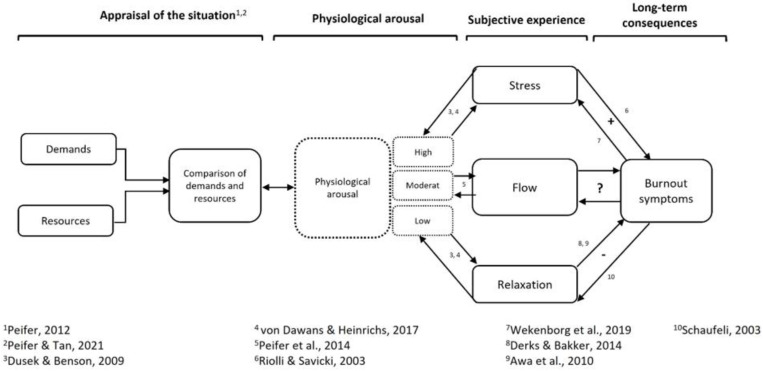
Initial Flow-Burnout-Model (adapted from Peifer & Tan [39], Peifer [42]). + = positive association; - = negative association.

**Figure 2 ijerph-19-03865-f002:**
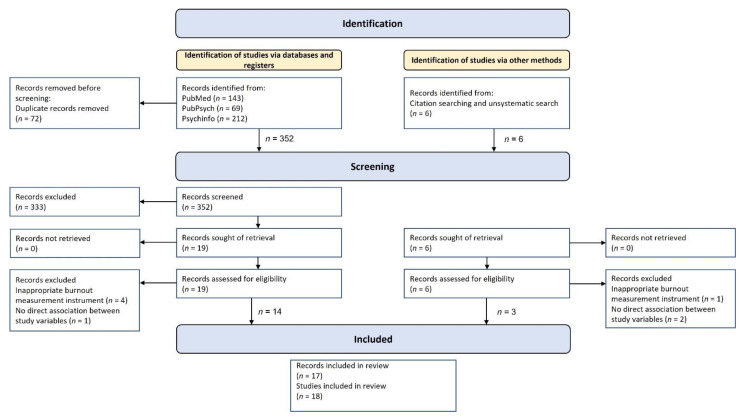
Selection process for the systematic review adapted from the PRISMA guidelines; Page et al., [112]).

**Figure 3 ijerph-19-03865-f003:**
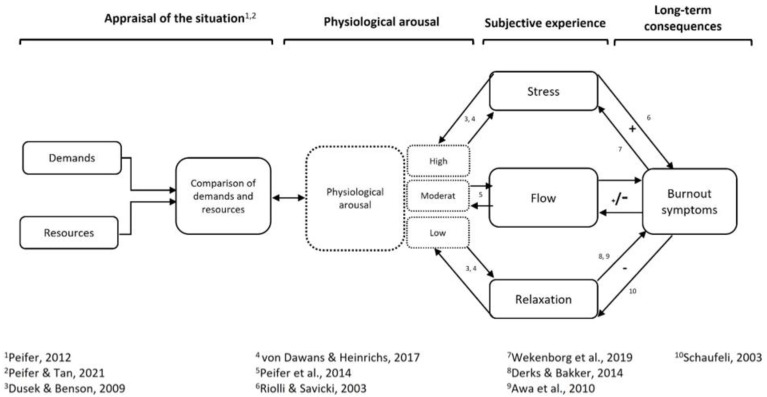
Adapted Flow-Burnout-Model (adapted from Peifer & Tan [39], Peifer [42]). + = positive association; - = negative association.

**Table 1 ijerph-19-03865-t001:** Overview of the results of the systematic review regarding design, country, sample, assessment instruments, and associations between flow and burnout symptoms.

Source	Design	Country	Sample	Assessment Instrument		Associations between Flow and Burnout Symptoms
				Flow Experience	Burnout Symptoms	
Bakker & Geurts, 2004 [77]	C-S	Netherlands	N = 1090 (*n*_1_ = 507; *n*_2_ = 202; *n*_3_ = 381) Sex ○ Female: 419 (*n*_1_ = 173; *n*_2_ = 126; *n*_3_ = 120) ○ Male: 671 (*n*_1_ = 334; *n*_2_ = 76; *n*_3_ = 261) Age ○ *n*_1_ = 35 (*SD* = 9.5) ○ *n*_2_ = 40 (*SD* = 9.0) ○ *n*_3_ = 40 (*SD* = 10.4) Context ○ Pension fund company (*n*_1_) ○ Occupational health services company (*n*_2_) ○ Insurance company (*n*_3_)	UWES [78] ○ AB (6 items)	MBI-GS [79] ○ EE (5 items)	**Correlation** *r*(AB; EE) = −0.16 ***
Baumgarten et al., 2020 [80]	C-S	France	N = 243 Sex ○ Female: 64 ○ Male: 179 Age ○ NR Context ○ residents (*n* = 141) neurosurgeons (*n* = 102)	WOLF [81]	MBI [69]	**Stepwise multiple regression** EE (Y) ○ AB (X): β = 0.13 * ○ EN (X): β = −0.30 *** ○ IM (X): n.s. CY (Y) ○ AB (X): n.s. ○ EN (X): β = −0.25 *** ○ IM (X): n.s. RPA (Y) ○ AB (X): n.s. ○ EN (X): β = −0.19 ** ○ IM (X): n.s.
Demerouti et al., 2012 [82]	LI (diary study—4 days)	Germany and Netherlands	N = 83 Sex ○ Female: 49 ○ Male: 34 Age ○ 41.86 (*SD* = 13.80) Context ○ Employees from 13 different organizations	WOLF [81,83] ○ 3 items per subscale	MBI-GS [79] ○ General EE (5 items) ○ EE at work/at bedtime (3 adapted items)	**Correlation** *r*(AB; general EE) = −0.12 * *r*(AB; EE at work)= −0.14 * *r*(AB; EE at bedtime) = −0.05 *r*(EN; general EE) = −0.43 ** *r*(EN; EE at work)= −0.44 ** *r*(EN; EE at bedtime) = −0.37 ** *r*(IM; general EE) = −0.13 * *r*(IM; EE at work)= −0.10 *r*(IM; EE at bedtime) = −0.03 **Multilevel estimates** EE at work (Y) ○ AB (X): Estimate = −0.07 ○ EN (X): Estimate = −0.19 * ○ IM (X): Estimate = −0.07 EE at bedtime (Y) ○ AB (X): Estimate = −0.05 ○ EN (X): Estimate = −0.17 * ○ IM (X): Estimate = −0.13
Kasa & Hassan, 2015 ^a^ [84]	C-S	Malaysia	N = 293 Sex ○ Female: 99 ○ Male: 194 Age ○ majority (52.6%) between 20 and 29 Context ○ Hotel employees	WOLF [81]	OLBI [85] ○ adapted version	**Correlation** *r*(flow, burnout symptoms): n.s.
Kasa & Hassan, 2016 [86]	C-S	Malaysia	N = 317 Sex ○ Female: 166 ○ Male: 151 Age ○ majority (70.7%) between 18 and 29 Context ○ Hotel employees	WOLF [81]	OLBI [85] ○ adapted version	**Structural model** flow and burnout symptoms: β = 1.46 *
Kasa & Hassan, 2019^a^ [87]	C-S	Malaysia	N = 293 Sex ○ Female: 99 ○ Male: 194 Age ○ 21–29 (53%) ○ 30–39 (30%) ○ 40–49 (16%) Context ○ Hotel employees	WOLF [81]	OLBI [88]	**Regression** flow (Y) ○ burnout symptoms (X): β = 0.08
Lavigne et al., 2012 [89] (Study 1)	C-S	Canada	N = 113 Sex ○ Female: 80 ○ Male: 33 Age ○ 29.43 (*SD* = 4.04) Context ○ Québec’s public service association	Flow experience at work scale [90] ○ adapted from Jackson & Marsh [91]	MBI—French version [92]	**Correlation** *r*(flow; EE) = −0.10 *r*(flow; CY)= −0.44 *** *r*(flow; RPA) = −0.60 ***
Lavigne et al., 2012 [89] (Study 2)	LI (T1 and after 6 months T2)	Canada	N = 325 Sex ○ Female: 172 ○ Male: 153 Age ○ 44.8 (*SD* = 9.64) Context ○ Professionals for the Québec government	Flow experience at work scale [90] ○ adapted from Jackson & Marsh [91]	MBI—French version [92]	**Correlation** *r*(flow (t1); EE (t1)) = −0.42 *** *r*(flow (t1); EE (t2)) = −0.40 *** *r*(flow (t1); CY (t1)) = −0.53 *** *r*(flow (t1); CY (t2)) = −0.54 *** *r*(flow (t1); RPA (t1)) = −0.66 *** *r*(flow (t1); RPA (t2)) = −0.51 *** *r*(flow (t2); EE (t1)) = −0.39 *** *r*(flow (t2); EE (t2)) = −0.46 *** *r*(flow (t2); CY (t1)) = −0.52 *** *r*(flow (t2); CY (t2)) = − 0.61 *** *r*(flow (t2); RPA (t1)) = −0.52 *** *r*(flow (t2); RPA (t2)) = −0.63 ***
Ljubin-Golub et al., 2020 [93]	C-S	Croatia	N = 213 Sex ○ Female: 149 ○ Male: 63 ○ 1 data for gender missing Age ○ 20.32 (*SD* = 2.16) Context ○ Students	WOLF-S [94]	OLBI-S [70,85] ○ adapted Croatian Version	**Correlation** *r*(AB; burnout symptoms) = −0.53 ** *r*(EN; burnout symptoms) = −0.60 ** *r*(IM; burnout symptoms) = −0.50 **
Mäkikangas et al., 2010 [95]	LI (T1, T2, T3 with 6 weeks in between)	Netherlands	N = 335 Sex ○ Female: 235 ○ Male: 100 Age ○ 30 (*SD* = 6.0) Context ○ Employees of an employment agency	WOLF [81]	MBI-GS [79] ○ Dutch version ○ EE (5 items)	**Correlation** *r*(flow (t1); EE) = −0.34 *** *r*(flow (t2); EE) = −0.34 *** *r*(flow (t3); EE) = −0.32 ***
Martínez-Zaragoza et al., 2014 [96]	C-S	Spain	N = 127 Sex ○ Female: 73 ○ Male: 54 Age ○ 42.41 (*SD* = 9.41) Context ○ Physicians	Flow Trait Scale-2 [91] in combination with PA Spanish adaption	MBI-GS [79] ○ translated and adapted by Salanova et al. [97]	**Correlation** *r*(flow + PA; burnout symptoms) = −0.08
Martínez-Zaragoza et al., 2017 [98]	C-S	Spain	N = 282 Sex ○ Female: 241 ○ Male: 41 Age ○ 36.49 (*SD* = 8.95) Context ○ Registered nurses	Flow Trait Scale-2 [91] ○ Spanish adaption	MBI-GS [79] ○ translated and adapted by Salanova et al. [97]	**Correlation** *r*(flow; EE) = −0.05 *r*(flow, CY) = −0.08 *r*(flow; RPA) = −0.32 **
Mosing et al., 2018 [99]	C-S	Sweden	N = 10.120 Sex: NR Age ○ 40.7 (*SD* = 7.75) Context ○ Swedish Twin Registry	SFPQ [100]	MBI-GS [79] ○ EE (5 items)	**Correlation** *r*(flow-work; EE) = −0.36 *** *r*(flow-maintenance, EE) = −0.24 *** *r*(flow-leisure; EE) = −0.23 *** *r*(flow-global; EE) = −0.34 *** *r*(flow-music; EE) = −0.03
Rodríguez-Sánchez et al., 2011 [101]	LI (ESM—14 days)	Netherlands	N = 100 (40 healthy vs. 60 burned-out) Sex ○ Female: 59 (*n*_healthy_ = 26; *n*_burned-out_ = 33) ○ Male: 41 (*n*_healthy_ = 14; *n*_burned-out_ = 27) Age ○ Healthy: 41.8 (*SD* = 10.0) ○ Burned-Out: 42.9 (*SD* = 8.8) Context ○ Healthy: different occupational groups ○ Burned-out: Dutch centers of expertise in burnout treatment	UWES [78] ○ 2 Items	MBI-GS—Dutch version [102]	**Correlation** *r*(flow; group) = −0.29 ** (person-level) *r*(AB; group) = −0.23 * (person-level) *r*(EN; group) = −0.30 ** (person-level) *r*(flow; group) = −0.13 **(time level) *r*(AB; group) = −0.11 ** (time level) *r*(EN; group) = −0.14 ** (time level) **Anova healthy vs. burned-out** flow: *t* = 8.70, *p* < 0.01 AB: *t* = 5.68, *p* < 0.05 EN: *t* = 9.62, *p* < 0.05 **Multilevel model** flow (Y) ○ Group (X): Estimate = −0.36 ** AB (Y) ○ Group (X): Estimate = −0.26 * EN (Y) ○ Group (X): Estimate = −0.45 ***
Schiefele et al., 2013 [103]	C-S	Germany	N = 281 Sex ○ Female: 197 ○ Male: 80 ○ 4 data for gender missing Age ○ 47.60 (*SD* = 9.82) Context ○ Teachers at different school forms	Combination of Flow-Short-Scale [104] and scale by Schiefele and Roussakis [105]	MBI—German version [106]	**Correlation** *r_s_*(flow; EE) = −0.14 * *r_s_*(flow; CY) = −0.24 ** *r_s_*(flow; RPA) = −0.24 **
Xanthopoulou et al., 2018 [107]	LI (diary study—5 days)	Netherlands and Poland	N = 50 Sex ○ Female: 46 ○ Male: 4 Age ○ 44 (*SD* = 11.8) Context ○ Various emotionally demanding, occupational contexts	WOLF [81] ○ 10 items ○ adapted version	MBI-GS [79] ○ EE (4 items)	**Correlation** *r*(flow; EE) = −0.52 ** (person-level) *r*(flow; EE) = −0.43 ** (day-level)
Xie et al., 2019 [108]	C-S	China	N = 1977 Sex ○ Female: 1407 ○ Male: 570 Age ○ 19.90 (*SD* = 1.67) Context ○ Medical students	WOLF [81] ○ modified Chinese version	MBI-ES [76] ○ Chinese version	**Correlation** *r*(flow; burnout symptoms)= −0.59 **
Zito et al., 2016 [109]	C-S	Italy	N = 279 Sex ○ Female = 201 ○ Male = 78 Age ○ 42 (*SD* = 8.56) Context ○ Nurses	WOLF [81] ○ translated by Colombo et al., [110]	OLBI [85] ○ EX (8 items)	**Correlation** *r*(flow; EX) = −0.54 **

Note. C-S = cross-sectional, LI = longitudinal, ESM = Experience Sampling Method, N = sample size, *SD* = Standard deviation, WOLF = Work-related Flow Inventory, WOLF-S = WOLF-Study Questionnaire, UWES = Utrecht Work Engagement Scale, SFPQ = Swedish Flow Proneness Questionnaire, MBI = Maslach Burnout Inventory, MBI-GS = Maslach Burnout Inventory—General Survey, MBI-ES = Maslach Burnout Inventory—Educators’ Survey, OLBI = Oldenburg Burnout Inventory, OLBI-S = Oldenburg Burnout Inventory –College Students, EE = emotional exhaustion, CY = cynicism, RPA = reduced personal accomplishment, PA = personal accomplishment, EX = exhaustion, AB = absorption, EN = enjoyment, IM = intrinsic motivation, T = timepoint, X = independent variable, Y = dependent variable, n.s. = not significant (statistical parameters not given), NR = not reported. * *p* < 0.05; ** *p* < 0.01; *** *p* < 0.001. For reasons of clarity and comprehensibility, the labels of the various subscales were harmonized and reversed recoded where appropriate. ^a^ The studies may have been based on the same data.

**Table 2 ijerph-19-03865-t002:** Overview of the positive, negative, and non-significant associations found in the studies.

Relationship	Flow	Burnout Symptoms			
		Total Score	EE/EX	CY	RPA
Positive association	Total score	1			
	AB		1		
	EN				
	IM				
Negative association	Total score	2	6	3	4
	AB	2	2		
	EN	2	2	1	1
	IM	1	1		
Non-significant association	Total score	3	3	1	
	AB		1	1	1
	EN				
	IM		2	1	1

Note. EE = emotional exhaustion, EX = exhaustion, CY = cynicism, RPA = reduced personal accomplishment, AB = absorption, EN = enjoyment, IM = intrinsic motivation. The results from Table 1 are summarized. Further associations with other influential factors are not considered.

## Data Availability

The data presented in this study are available on request from the corresponding author.

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
