# Peer review of "The Relationship between Flow Experience and Burnout Symptoms: A Systematic Review"

_ijerph, 2022, doi:10.3390/ijerph19073865_

Round 1
Reviewer 1 Report
Thanks for the opportunity to read the article “The relationship between flow experience and burnout symptoms: A systematic review”. Based on the analysis of the literature the Authors developed a psycho-physiological “Flow-Burnout-Model”, which postulates either positive or negative associations between flow and burnout symptoms, depending on the prevailing situational and personal conditions. A strength of the paper is the large and diverse literature review and empirical studies review focused on the relationship between flow and burnout. It also highlights the importance of considering the cultural context of the studies (the presented studies were carried out in ten different countries). The reviewed article is cognitively interesting and well-written.
I have only one recommendation for improving the article. In the context of choosing the variable of “burnout”, it would be advisable to reference to conceptualisation taking into account four dimension constitute the core burnout: exhaustion, emotional impairment, cognitive impairment, mental distance and secondary symptoms: psychological distress, psychosomatic complaints. See: Schaufeli, W., Desart, S., De Witte, H. (2020). Burnout Assemssment Tool (BAT) – Development, Validity, and Reliability. International Journal of Environmental Research and Public Health, 17, 9495; doi:10.3390/ijerph17249495.It would be worthwhile to present a broader justification for the adopted definition of burnout in the article.
The application of methodology approach to the analysis of issues undertaken in the article is justified and appropriate.
Reviewer 2 Report
Dear authors,
Thank you very much for the opportunity to read your manuscript.
I will suggest some points in order to help and be food for thought.
The first comment is that the work is well organised and well written, which allowed us to go deep into the content.
The first point that calls attention is a flow-burnout model as JDR is a model that in some sense predicts this relationship if we understand flow as part of the motivation or at least share some dimensions. Also, as the authors found that burnout and flow are negatively related, the motivation-burnout part of JDR model posits it. If I understand well, the proposition was to unpack the subjective experience that people once felt D-R balanced or unbalanced. JDR, as reviewed by Bakker and Demerouti 2017, also includes personal demands and resources as coping strategies.
It does not invalidate the work and the proposition, but from my point of view is not a new model; it is the unpack of a part of the model that already exists.
Another point about your preventive proposition is that flow is “post-fact”, it is impossible to induce flow. It is possible to understand that it is protective, but it is only protective because people already have a positive circumstance. As a psychological state, the worker only feels it.
There are pieces of evidence between burnout and recovery relationship.
Why do authors only look for research in English? Non-studies from Africa and/or Latin America were included. Authors comment that the results were different in Malaysia, so including other results from other collectivist countries could be crucial.
The search items only included the umbrella of burnout and flow, but not burnout-relaxation and burnout-stress (or, in some sense, the model covers stress as a predictor of stress which is odd).
How Table 1 is organised? It is not clear, but it seems by year.
Another point is to think about the type of task of the sample as a moderator or if the job characteristics would influence the flow experience. Authors could organise or include a description of the samples as “blue collar”, White, pink or green (i.e. army or police). As some tasks are known as more exhaustive than others, it could help to understand the results.
Another suggestion for the authors is to include a 3x3 table including positive, negative and null relation, in the sense that authors could find, as Peiró suggests, synergic relation, antagonist relation and null relation between variables and this also shows possible gaps in the literature.
The explanation offered by the articles found of null and antagonistic relations found on the articles are not systematised by the authors and could also be interesting, mainly if the idea is to develop a new model.
Hopefully, the comments will be helpful.
Reviewer 3 Report
Comments:
I hope all these suggestions will not discourage the author(s) keep working on this manuscript in the future. Good luck to the author(s).
Additional Questions:
- Originality: The study of The relationship between Flow experience and burnout symptoms: A systematic review sinificantly contributes to the understanding of trends in Flow and burnout studies and relationships between the variables and methodologies and expands our scientific perspective. In that sense, the research context of this paper is very interesting. In addition, the area of the research highlight the increase in burnout and flow, which makes it a relevant issue nowadays. Thus, I believe further research in this topic is needed. Nevertheless, there are some relevant concerns in this paper which are shown below.
- In general terms, the paper an understanding of the relevant literature in the labour market field, but I do not see the introduction. In your introduction you might bring up many important points, however you do not really link them together and the text just flatters around. Here you have to write a stringent storyline guiding the reader. I can not see the point of your article clear, I do not see if there are other articles about this topic and most important I do not see the gap in the science that this paper cover.
- Concretly, in the introduction in line “While some people seem to handle challeging situations easily, others are unable to cope with high job demands and consequently suffer from burnout symptoms” is the main point of this article which is your main contribution and it must be something more important than this line, because it might be the gap.
- Relationship to Literature In general terms, the paper demostrate an understanding of the relevant literature in the labour market field. Thus, I think the literature review is very rich in this paper. However, there are some relevant concerns in this section that should be improved:
- Some important statements should be supported by the literature. I think there are literatura review in the introduction which should be moved.
- Besides in line 59 I can see a full stop which must be remove “in light of the severe consequences of burnout symptons. Among them impaired life satisfaction.
- I think is not neccessary the point 2.2. Physcal arousal becase is so medical for this area of expertise.
- ¿Figure 1 and figure 3 is the same? It is repeated and it must be remove it.
- There are errors in nominations like Associations between Flow and burnour sysptoms measured with MBI, because its is 2.3 but it might be 4.2.1 and so on.
- Implications for research, practice and/or society: The author does not explain what the theoretical contributions are or the practical implications for the topic. Neither makes proposals or recommendations. The conclusions are not solidly based in the empirical obtained results.
- Concluions:
Authors did not formulate Conclusions that are anchored in the research results. The text of the part contains NO conclusions. The text should be added with elements that are anchored in research findings. The text should contain analytical statements describing the MERIT TOPIC. The further research recommendation should (might) be included.
The possible solution for the Conclusions part is dividing it into three parts:
- Conclusions resulting from the STATE OF ART analysis,
- Conclusions resulting from own research, but only from the study described in the current text,
- Policy implications,
- Further research directions.
Conclusions should be fully anchored in the results of the methodological proposals and analysis, drawn from the empirical material. The last part should be formulated based on the GOAL formulation, i.e.
- WHY the findings are essential;
- for WHOM and
- WHAT FOR; for which type of policy or for what kind of decisions.
Statements may be complemented with the information on
- HOW the results were obtained and
- HOW the results may be implemented in the education service providing system.
Round 2
Reviewer 2 Report
Dear authors,
Thank you for the opportunity to reread your work. In general, the manuscript improves. Although the Table 2 notes are not in order, which I suggest reviewing, added an important contribution to the work. The insertions were correct. I´m still puzzled how those databases are biased and do not include any African or Latin-American country.
Thank you very much, congratulations.
Reviewer 3 Report
Congratulations great job!